# Assessing the Impact of School Rules and Regulations on Students' Perception Toward Promoting Good Behavior: Sabian Secondary School, Dire Dawa, Ethiopia

**Alemneh Amesalu Fekadu**

Department of Statistics, College of Natural and Computational Sciences, Dire Dawa University, 1362 Dire Dawa, Ethiopia; amsalu004@gmail.com; Tel.: +25-190-962-6781

**Abstract:** Discipline is an important component of human behavior, and one could assert that without it, an organization cannot function well toward the achievement of its goals. The aim of this study was to assess the impact of school rules and regulations on students' perception toward promoting good behavior. The data were obtained from 438 respondents through a mailed questionnaire instrument. The data were tabulated, and Pearson's chi-square test was applied for inferential analysis. Around 33.1% of the students had a negative perception of school rules and regulations about promoting good behavior, whereas 66.9% of them had a positive perception. A $p$-value of 0.015 (<5% significance level) indicated that there is a significant association between students' awareness on school rules and regulations and their perception toward promoting good behavior. Students' attitudes on school rules and regulations and perception toward promoting good behavior were statistically associated at a $p$-value of 0.012. Parents' educational levels had a significant effect on students' perception toward promoting good behavior. Generally, students' awareness on school rules and regulations, parents' education levels, civics and ethical education scores, and students' attitudes toward promoting good behavior were found as significant effects on perception toward promoting good behavior.

**Keywords:** rules; regulations; promoting good behavior; Pearson's Chi-square test; SPSS

## 1. Introduction

### 1.1. Background of the Problem

Students are key stakeholders and the most essential resources in education. It is absolutely necessary to direct students to exhibit an acceptable attitude and behavior within and outside the school. In an attempt to achieve an organized and peaceful school environment and maintain law and order, the school management specifies rules and regulations to guide the activities of members of the educational institution [1].

Education is a lifelong process that begins at home and continues as a formal and informal training. How a person learns and his or her success in learning the lessons of life are contingent upon earlier lessons. Attitude affects perceptions and development through the learning process and is ultimately manifested in behavior [2].

Different strategies to maintain discipline have been employed both at the national and school levels. Among others, the adoption and deployment of school rules and regulations in secondary schools is the main target for monitoring and curbing students' behaviors [2].

Discipline is a prerequisite for almost everything a school can offer to students [3]. It links both the culture and the climate of a school, and for a satisfactory climate to exist in a school, a certain level of discipline must exist. In schools where discipline is a serious problem—for example, when students bully each other—parents prefer to transfer their children to better schools. Since the well behaved students usually perform well, their transfer to another school can affect the overall performance of the former school.

Perception is a process of interpreting and understanding one's environment. It is the process through which people select, organize, and interpret what they see, hear, touch, and smell, and create meaning from such things and respond to the world around them. Furthermore, perception is also a process through which people receive, organize, and interpret information from their environment. The secondary school students' perception on school rules and regulations, therefore, has a great effect on the way they obey and adhere to such rules and regulations [4].

### 1.2. Motivation and Innovation

Some students do not abide by the school rules and regulations, and for the same reason, they dropout from school. Some students are not aware of and have less perception about school rules and regulations that can instill in them acceptable behavior. Schools play an important role in the socialization process of young people; it is where they learn to regulate their own conduct, respect others, manage their time responsibly, and thus become responsible citizens [5]. In this backdrop, this study aims to assess the impact of school rules and regulations on students' perception toward promoting good behavior, considering a case study of Sabian Secondary School, Dire Dawa, Ethiopia.

### 1.3. Beneficiaries

The following stakeholders are expected to be benefited:

- It will address the issues for curriculum developers related to students' discipline in the context of creating good citizens.
- It will help school administrators to come up with rules and regulations that are effective and efficient in discipline enhancement among students and staff.
- It will help in addressing issues of student indiscipline and parents, and the wider society will benefit since the costs associated with student indiscipline will be minimized.

Besides this, this study expects to promote further research on the relationship between the school rules and regulations and a good citizen. In other words, this study aims to develop insight for critically assessing the content of school rules and recommendations that can produce quality citizenry.

### 1.4. Operational Definitions

- School rules: These include principles or orders that guide the behavior of students in schools.
- School regulations: These include daily timetable procedures of carrying out activities in school as individuals or in groups.

## 2. Literature Review

### 2.1. School Rules and Punishing Offenders

In a proper learning situation, a disciplined student is the one expected to do the right things at the right time [6]. A disciplined student is the one who is in the right place at the right time. Punishment is often used for students who break school rules or do not follow school regulations [7].

According to [8], students realize that punishments are an effective method of remediating individual misbehavior and therefore improving school order. Similarly, in the views of [9], students understand that punishment can be an effective way of controlling misbehavior when it is fair and consistent. It acts as a motivator for improving their learning and academic performance. Students believe that punishments in a school system are expected to teach them the relationship between their behaviors and outcome or accountability for their mistakes [10].

### 2.2. Students' Participation in the Formulation of School Rules and Regulations

Student participation can be defined as the level of engagement and intrinsic interest that a student shows in school rules and regulations. Student participation in the implementation of school rules refers to the work of students' representative bodies (school councils, student parliaments) in the formulation of school rules. It is also used to encompass all aspects of school life and decision making, where students can make a contribution informally through individual negotiation as well as formally through purposely created structures and mechanisms. It also refers to the participation of students in collective decision-making at a school or class level and to the dialogue between students and other decision makers. It is not solely based on consultation or a survey among students [11].

### 2.3. Students' Attitude toward School Rules and Regulations

Most students think that many school rules are good and without these rules, the school would not be a pleasant place. As for the reason why these school rules exist, students primarily and most frequently explain that they are there to prevent students from harming or hurting others or from making other students upset, unhappy, frightened, or feel left out [11].

## 3. Data and Methodology

Study Area: Dire Dawa Administration is located in the eastern part of Ethiopia, 515 kilometers from Addis Ababa, the capital city of Ethiopia, and bounded by Oromia Regional State in the south and by Somali Regional State in the north, east, and west. Dire Dawa Administration is characterized by relatively high temperatures throughout the year with minor seasonal variations.

### 3.1. Target Population

The target population was from Sabian Secondary School in Dire Dawa city. The study included grades nine, ten, eleven, and twelve, and the total number was 2520 for the secondary school students in the 2017/2018 academic year.

### 3.2. Sampling Techniques and Data Collection Instrument

The sampling technique used in this study was stratified based on the grade level. These levels were first stratum (Grade–9), second stratum (Grade–10), third stratum (Grade–11), and fourth stratum (Grade–12), and simple random sampling technique was employed to select the sampling units from each stratum. Data were collected using a self-administered questionnaire instrument. The questionnaire was prepared in English and translated into the local language, Amharic, for more clarification on questions and to improve the quality of the data.

### 3.3. Sample Size Determination

The appropriate sample size depends on available resources and the level of precision required. Our sample size was calculated using Cochran's (1977) sample size calculation formula:

$$n_0 = \frac{(Z_{\frac{\alpha}{2}})^2}{d^2} P(1 - P). \tag{1}$$

If $\frac{n_0}{N} > 5\%$; then, $n = \frac{n_0}{1 + \frac{n_0}{N}}$ and if $\frac{n_0}{N} \leq 5\%$, $n = n_0$.

where, $P = 0.69$ is the proportion of students with a positive perception on school rules and regulations regarding promoting good behavior, taken from Asnake and Ashenafi (2018), and $q = 1 - P = 0.31$ is the proportion of students with negative perception on school rules and regulations toward promoting good behavior, $z = 1.96$ is the critical values of standard normal cumulative distribution that corresponds to $\frac{\alpha}{2}$ with 5% significant level, $N$ = the total number of students in Sabian school (2520), and $d = 0.04$ is the maximum allowable error the researcher will tolerate.

$$n_0 = \frac{(1.96)^2(0.69)(0.31)}{0.04^2} = 514,$$
$$n = \frac{n_0}{1 + \frac{n_0}{N}} = \frac{514}{1 + \frac{514}{2520}} = 426.91 \approx 427. \tag{2}$$

By adding 5% contingency for the expected non-response rate, the final sample size estimated is:

$$n = n + 0.05n = 427 + 0.05(427) = 448.35 \approx 449. \tag{3}$$

Proportional allocation: the sample size for each stratum (grade level) was calculated as follows: $n_h = \frac{N_h}{N} n$ and is given in Table 1.

**Table 1.** Summary of proportional sample size allocation.

| Grade Level | Number of Student ($N_h$) | Sample Size ($n_h$) |
|---|---|---|
| Nine | 937 | $\frac{937}{2520} * 449 = 166.95 \approx 167$ |
| Ten | 805 | $\frac{805}{2520} * 449 = 143.43 \approx 143$ |
| Eleven | 459 | $\frac{459}{2520} * 449 = 81.78 \approx 82$ |
| Twelve | 319 | $\frac{319}{2520} * 449 = 56.83 \approx 57$ |
| Total | 2520 | 449 (Total sample size) |

Dependent variable: Students' perception on school rules and regulations toward promoting good behavior ($Y_i$). It is dichotomous, and when a student has positive perception on the school rules and regulations on promoting of good behavior, it is $Y_i(1)$ or otherwise $Y_i(0)$.

Independent variables: Students' attitude towards schools rules and regulations, awareness on school rules and regulations, participation in the formulation or implementation of schools rules and regulations, and civics and ethical education score, parents' education levels, and parents' employment status.

*3.4. Statistical Model*

Statistical models such as descriptive statistics (graphs and tables for aggregation of the collected data and determination of patterns) and Pearson's chi-square test were employed to analyze the collected data. Statistical Package for Social Sciences (SPSS version 20.0) was used for data analysis.

Chi-Square Test of Independence

The chi-square test of independence was applied for two categorical variables to examine the significant association between them. In order to apply the chi-square test, a null hypothesis was required. Therefore, the null hypothesis was that "there is no relationship between the students' perception toward promoting good behavior and a single independent variable", whereas an alternative hypothesis was the opposite of the null hypothesis. The resulting decision was that if the *p*-value was less than 5%, there was a relationship between the two variables.

$$\text{Chi} - \text{Square } X^2 = \sum \left[ \frac{(O_i - E_i)^2}{E_i} \right], \; X^2((R\text{--}1) * (C\text{--}1)); \; E_i = \left( \frac{row\ count\ total * colomun\ count\ total}{total\ counts} \right)$$

where $O_i$ is the observed number of cases in category *i*, and $E_i$ is the expected number of cases in category i; R and C are the total number of rows and columns, respectively, and (R–1)*(C–1) is the degrees of freedom. The *p*-value indicates the probability of chi-square statistics and its degrees of freedom.

## 4. Results and Discussion

*4.1. Descriptive Statistics*

4.1.1. Perception on the School Rules and Regulations toward Promoting Good Behavior

The total sample size was 449, but from those, 11 respondents' questionnaires were rejected due to incomplete responses, thus the final sample included 438 respondents. About 33.1% of the students negatively perceived school rules and regulations regarding the promotion of good behavior, whereas 66.9% of them perceived them positively (Table 2).

**Table 2.** Perception on school rules and regulations in promoting good behavior.

| Perception on School Rule and Regulation in Promoting Good Behavior | Frequency | Percent | Valid Percent | Cumulative Percent |
|---|---|---|---|---|
| negative perception | 145 | 33.1 | 33.1 | 33.1 |
| positive perception | 293 | 66.9 | 66.9 | 100.0 |
| Total | 438 | 100.0 | 100.0 | |

4.1.2. Awareness on School Rules and Regulations and Perception in Promoting Good Behavior

About 70.2% of students had awareness or more understanding of the school rules and regulations toward promoting good behavior, whereas 29.8% of them perceived them negatively. Among them, 57.8% of students had awareness and perceived school rules and regulation positively, while 42.2% perceived them negatively (Table 3).

Participating in school rules and the formulation or implementation and perception in promoting good behavior:

Around 68.9% of the students were participating in school rule formulation or implementation and had a positive perception toward creating good behavior. The remaining 31.1% perceived it negatively. Of the respondents, 61.8% were involved in the formulation or the implementation of school rules and had a positive perception in the promotion of good behavior, while 38.2% of them perceived it negatively. Nowadays, at different school levels, many student representatives are involved in the implementation of school rules and regulations. Representatives play a great role in fostering a conducive teaching and learning environment that promotes well-behaved students.

### 4.1.3. Parents' Education Level and Perception toward Promoting Good Behavior

Father's education level: Around 51.7% of the respondents from uneducated fathers positively perceived school rules and regulation toward promoting good behavior, while 48.3% of them perceived them negatively. It was found that 66.7% of students from primary educated fathers perceived the school rules and regulations positively toward promoting good behavior, whereas 33.3% of them perceived them negatively. On the other hand, 71.8% of students from secondary school educated fathers positively perceived school rules and regulations toward promoting good behavior, and the remaining 28.2% perceived them negatively. Furthermore, 75.7% of respondents from diploma or above educated fathers were positively perceived school rules and regulations toward promoting good behavior, while 24.3% of them had a negative perception.

Mother's education level: Approximately 52.9% of the respondents from uneducated mothers had a positive perception of school rules and regulations toward promoting good behavior, while 47.1% of them perceived them negatively. On the other hand, 61.3% of students from a primary school educated mother positively perceived school rules and regulations toward promoting good behavior, and the remaining 38.7% perceived them negatively. Moreover, 69.5% of respondents from secondary educated mothers perceived school rules and regulations positively in promoting good behavior, while 30.5% of them perceived them negatively. Similarly, 75.2% of students from a diploma or degree educated mother were positive on school rules and regulations in promoting good behavior, and the remaining 24.8% of them were negative. Therefore, the proportion indicates that the parents' education levels have a positive impact on students' perception toward school rules and regulations in promoting good behavior. These data indicate that more educated parents will advise and teach their children about following school rules and regulations, which will promote good behavior.

### 4.1.4. Parents' Employment Status and Perception in Promoting Good Behavior

Father's employment status: Approximately 57.1% of respondents from fathers with no occupation positively perceived school rules and regulations toward promoting good behavior, whereas 42.9% of them perceived them negatively. On the other hand, 68.2% of respondents from self-worker fathers positively perceived school rules and regulations toward promoting good behavior, whereas 31.8% of them perceived them negatively. Among the students from employed fathers, 68.1% had a positive perception of school rules and regulations toward promoting good behavior, while 31.9% of them negatively perceived them.

### 4.1.5. Punishment for Misbehavior and Perception in Promoting Good Behavior

Of the respondents, 61.6% disagreed that punishment encourages following school rules and regulation and perceived it positively toward promoting good behavior, whereas 38.4% of them perceived it negatively. On the other hand, 70.1% of respondents agreed that punishment encourages following school rules and regulations and perceived it positively toward promoting good behavior, while the remaining 29.9% perceived it negatively.

**Table 3.** Summary statistics and chi-square results for perception and explanatory variables.

| Explanatory Variable Categories | | Perception on School Rule and Regulation in Promoting Good Behavior | | | | Chi-Square (*p*-Value) (dof) |
|---|---|---|---|---|---|---|
| | | Negative perception | | Positive perception | | |
| | | Count | Percent (%) | Count | Percent (%) | |
| Awareness on school rules and regulations | No | 49 | 42.2% | 67 | 57.8% | 5.95 (0.015) (1) |
| | Yes | 96 | 29.8% | 226 | 70.2% | |
| Participating in school rules formulation or implementation | No | 47 | 38.2% | 76 | 61.8% | 2.01 (0.156) (1) |
| | Yes | 98 | 31.1% | 217 | 68.9% | |
| Father's education level | Uneducated | 56 | 48.3% | 60 | 51.7% | 14.73 (0.002) (3) |
| | Educated at primary school level | 23 | 33.3% | 46 | 66.7% | |
| | Educated at secondary school level | 33 | 28.2% | 84 | 71.8% | |
| | Having diploma and above | 33 | 24.3% | 103 | 75.7% | |
| Mother's education level | Uneducated | 49 | 47.1% | 55 | 52.9% | 18.13 (0.000) (3) |
| | Educated at primary school level | 12 | 38.7% | 19 | 61.3% | |
| | Educated at secondary school level | 47 | 30.5% | 107 | 69.5% | |
| | Having diploma and above | 37 | 24.8% | 112 | 75.2% | |
| Civic and ethical education semester score | Less than 50 | 19 | 67.9% | 9 | 32.1% | 19.27 (0.000) (2) |
| | Between 50 and 70 | 77 | 34.4% | 147 | 65.6% | |
| | Greater than 70 | 49 | 26.3% | 137 | 73.7% | |
| Father's employment status | No occupation | 21 | 42.9% | 28 | 57.1% | 2.37 (0.306) (2) |
| | Self-worker | 55 | 31.8% | 118 | 68.2% | |
| | Employed | 69 | 31.9% | 147 | 68.1% | |
| Mother's employment status | House wife | 48 | 43.6% | 62 | 56.4% | 9.45 (0.009) (2) |
| | Self-worker | 45 | 26.0% | 128 | 74.0% | |
| | Employed | 52 | 33.5% | 103 | 66.5% | |
| Punishment encourages the following rules and regulations. | Disagree | 63 | 38.4% | 101 | 61.6% | 3.34 (0.68) (1) |
| | Agree | 82 | 29.9% | 192 | 70.1% | |
| Attitude toward school rules and regulations | Negative attitude | 34 | 36.2% | 60 | 63.8% | 6.25 (0.012) (1) |
| | Positive attitude | 111 | 32.3% | 233 | 67.7% | |

## 4.2. Test of Associations (Chi-Square)

This section describes testing the null hypothesis. In other words, we determined if there existed a relationship between the perception on school rules and regulations toward promoting good behavior and the different categorical explanatory (independent) variables by using the chi-square test.

Awareness of school rules and regulations: It can be seen in Table 3 that the chi-square value between students' awareness of school rules and regulations and the perception toward promoting good behavior was 5.95 (*p*-value 0.015). This indicates that there is a significant association between students' awareness of school rules and regulations and their perception toward promoting good behavior at a 5% level of significance. The frequency distribution results also show that there is a significant difference between the students who have awareness of school rules and regulations relative to those who have no such understanding of it toward promoting good behavior.

Participation in school rule formulation or implementation: The chi-square value between participating in school rule formulation or implementation and perception toward promoting good behavior was 2.01 (*p*-value 0.156). Since the *p*-value of 0.156 was greater than the 5% level of significance, we have evidence to say that there is no statistical association between participating in school rules formulation or implementation and perception toward promoting good behavior. This indicates that participating in school rule formulation or implementation does not affect promotion of good behavior.

Father's education level: The chi-square and *p*-value between the father's education level and students' perception toward promoting good behavior were 14.73 and 0.002, respectively. Since the *p*-value was less than a 5% significance level, there exists a relationship between students' perception toward promoting good behavior and their father's education level. Similarly, the frequency distribution reveals that students with positive perception toward promoting good behavior who have fathers with education levels with a diploma or above make the highest proportion in relation to others.

Mother's education level: The estimated chi-square and *p*-values between the mother's education level and students' perception toward promoting good behavior were 18.13 and 0.000, respectively. We can conclude that there is a statistically strong relationship between students' perception toward promoting good behavior and their mother's education level at a 5% level of significance. Additionally, the frequency distribution results show that the students from more educated mothers are the highest in having positive perception toward promoting good behavior as compared to students from low educated mothers.

Civics and ethical education score: The chi-square and *p*-values between civics and ethical education semester scores and students' perception toward promoting good behavior were 19.27 and 0.000, respectively. Since the *p*-value was less than 1% significance level, it can be said that there is a statistically strong relationship between civics and ethical education scores and students' perception toward promoting good behavior. The summary statistics also show that the students' perception toward promoting good behavior significantly varies with their scores in civics and ethical education.

Father's employment status: The estimated chi-square and *p*-values between students' perception toward promoting good behavior and father's employment status were 2.37 and 0.306, respectively. This indicates that there is no relationship between a father's employment status and students' perception toward promoting good behavior at a 5% level of significance. In other words, the father's employment status does not have any effect on students' perception toward promoting good behavior.

Mother's employment status: The estimated chi-square and *p*-values between students' perception toward promoting good behavior and the mother's employment status were 9.45 and 0.009, respectively. This indicates that there exists a relationship between the mother's employment status and students' perception toward promoting good behavior at a 5% level of significance.

Punishment encourages following rules and regulations: The chi-square and *p*-values between punishments encouraging following rules and regulations and perception toward promoting good behavior were 3.34 and 0.068, respectively. We have evidence to conclude that there is no statistical association between punishments encouraging following rules and regulations and promoting good behavior at a 5% level of significance.

Attitude toward school rules and regulations: The chi-square and *p*-values between students' attitudes toward school rules and regulations and their perception in promoting good behavior were 6.25 and 0.012, respectively. Therefore, we can assert that there is a statistically significant relationship between students' attitudes on school rules and regulations and their perception toward promoting good behavior at a 5% level of significance.

*4.3. Discussions*

The data were collected from 438 students with an aim to assess the impact of school rules and regulations on students' perception toward promoting good behavior. The participants were recruited from Sabian Secondary School at Dire Dawa, and the data were analyzed using descriptive statistics and inferential analysis, particularly by the chi-square test.

The results from descriptive statistics show that around 33.1% of the respondents had a negative perception of school rules and regulations toward promoting good behavior. Therefore, the concerned bodies should promote developing an understanding of secondary school rules and regulations to instill good behavior in students. This is because positive and responsible behavior by students is essential for the smooth running of the school, for the achievement of optimal learning opportunities, and for the development of a supportive and cooperative school environment.

This study shows that students' awareness of school rules and regulations and their perception toward promoting good behavior are statistically associated. This means that a better understanding of the school rules would increase the effectiveness of rules in promoting good behavior of students.

Students' attitudes on school rules and regulations have a significant association with the perception toward promoting good behavior. Students should have enough understanding about school rules and regulations to exercise easy and effective implementations. A study by [9] found that many students think that most school rules are good and without these rules, the school would not be a pleasant place.

Civics and ethical education scores show a significant impact on students' perception toward promoting good behavior. This means learning civics and ethical education plays an important role in creating a good learning environment.

## 5. Conclusions and Recommendations

*5.1. Conclusions*

The findings of this study show that around 33.1% of the students have a negative perception of school rules and regulations. This indicates that they do not adequately perceive school rules and regulations toward promoting good behavior.

The improvement in students' awareness of school rules and regulations would enhance their perception toward promoting good behavior because when students have enough understanding of school rules and regulations, they can behave well, and this offers a good learning environment.

Parents' educational level has a significant effect on students' perception toward promoting good behavior. More educated parents play a better role in enhancing and implementing school rules and regulations for promoting good behavior.

Civics and ethical education score is another significant factor in this direction, as low scorers in civics and ethical education are negatively associated with good behavior.

*5.2. Recommendations*

In order to prescribe the standards of good behavior, school rules and regulations should be emphasized, and enough awareness should be aroused in students. In future work, the effectiveness of school rules and regulations on academic achievement and the impact on life value should be investigated.

**Funding:** This work was not supported financially by any organization or individuals.

**Acknowledgments:** All participants involved in the study to provide relevant information and secondary school directors were duly acknowledged for their contributions.

**Conflicts of Interest:** The author declares that there is no conflict of interest regarding the publication of this paper.

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
