# Peer review of "Assessing the Impact of School Rules and Regulations on Students’ Perception Toward Promoting Good Behavior: Sabian Secondary School, Dire Dawa, Ethiopia"

_stats, doi:10.3390/stats2020015_

Reviewer 1 Report

The paper discussed the interesting topic of whether school rules and regulations matter on students' perception towards promoting good behavior. The findings are interesting and may be helpful in the management of secondary schools. However, I have some questions that may need to be answered:

On the top of pg. 4, 'q' was not used in the equation, but referenced and explained below, is q=1-p?

Why use the grade number n in the equation shown in line 143? This makes a larger weight to the higher grade to the lower grade. Shall the high grade have larger weight to fit the research goal?

Line 221, is the word 'reaming' a typo? remaining?

The last sentence of pg 6 did not mention the confidence level when you say statistically significant relationship, commonly we use 95%, or 99% of confidence level in such scenarios.

For easy reading, I would like to suggest you reformat the TABLE 3, e.g, use portrait layout and add lines between rows.

In section 4.2, the author only analyzed the relationships that are significant, it is also beneficial to emphasize those are not significant and if possible, analyze why (this could be difficult).

It is worth studying the relationship between multiple variables and the students' perception, e.g., father's education level+mather's education level V.S. students' perception. But, this could be a further work.

The meaning of magnitudes of chi-square values are related to their degrees of freedom (dof), do all chi-square test in your table have the same dof? It is worth indicating in some place of your table, if they have different dof, provides those dofs.

Author Response

Some review comments highlight responses 

q = 1-P =0.31 is the proportion of students having negative perception on school rules and regulation towards promoting good behavior

Line 143 why use grade number n in equation is 

Answer:

n is total sample size to be included in the study not grade number and it is estimated by using proportional allocation because sampling design used is stratified sampling technique staratified parameter is grade level (9, 10, 11&12)

shall high grade  have large weight?

Answer: no but large number of students will give us large sample size allocation and it leads good result to fit our research

line 221 : reaming = remaining

comment on page 6 the paragraph mentioned is totally misplaced and i want to delete it because it has mentioned on page 8 last paragraph and iconfidence level used is 95% i will add it on the correct place  

in section 4.2 

yes now all are included 

studying the relationship between peception towards and multiple response variable would be conducted by using logistic regression but now not

Generally detail comments were incorporated in the manuscript 

Reviewer 2 Report

The paper is well written and constructed.  Motivation, innovation and literature review are sufficient.  Authors declare the objective and the purpose of this paper  and the results presented analytically.  I believe that the paper should be accepted with minor revision.  It is very important that the authors have presented some of the results  in the abstract.

Comments

1)    In the abstract after the word background the first word must have the first capital letter: Discipline.

2)    Similarly after the word Method and Results the first word  must have the first capital letter.

3)    Authors should add some keywords such as Pearson Chi-Square Test, SPSS and statistically significant.

4)    In the last paragraph of the introduction the authors must  re-read the sentence Â«Similarly, Jones and George (2006)…to the world  around them. I think that the words touch smell should be separated by commas or after the comma authors should not use the word â€śand”  (touch smell, and taste and make).  Moreover authors are better to write Objective of the proposed work:  To access school rules and regulations…Dire Dawa.

5)    I believe that the title of subsection 1.2 Beneficiaries should  change. In these paragraphs authors declare clearly the motivation   and the innovation of the proposed work. The motivation and the   innovation are the most important components of a paper.  The new title of the subsection 1.2 should be   â€śMotivation and Innovation of the Proposed Work”

6)    In the first paragraph of Subsection 1.3 Operational definitions: I do not understand the meaning of the following proposal: “For example, all secondary students are expected to be in school uniforms when in school”.

7)    In the end of the second paragraph of Subsection 1.3: I think that the correct sentence is: School daily timetable and procedures of carrying out activities in school individually or in a group.

8)    At the end of third page authors should change the number of subsection. The correct number is 3.3 Sample size determination.

9)    At the beginning of fourth page authors should refer that (1-P) is equal to q (Series 132).  Moreover the output 514/2520>5% so authors chose the first formula (first if- Series 128 and 138)

10) How the results in Table 1 have been derived?  For example for grade level nine the output is (937/2520)*449=167  for grade level ten is (801/2520)*449=143.11) In Page 4 authors should declare some literature reviews about  the formulas. Otherwise, authors should present briefly Cochran’s  (1977) sample size calculation formula. The above observations  (Comments 9-11) are made because readers should not think about how  the calculations were made but should be presented to them.12) Because the paper is 10 pages, the authors could present  in 2 to 3 paragraphs the Pearson Chi-Square Test (before the results). Authors should provide some details about the calculation of the  Statistic Test (X2) (expected and observed values), the selection  of the level of the significance a=0.05 and confidence Interval Îł=95%, the calculation of degrees of freedom and P-Value.13) At the beginning of page 6 in father education level authors  should replace the 28.8 percent with 28.2 percent.  The 71.8 percent and the 28.8 percent is greater than 100%.14) In Table 3 i think that authors have used for the calculation of  chi-square test and P-Value the level of significance a=0.05.  Authors should be declaring it in the beginning of the fourth Chapter. According to Table 3 only 3 examined categorical explanatory variables (participating in school rules formulation or implementation (0.156),  father employment status (0.306) and punishment encourages following  rules and regulations (0.68)) are independent (no relationship) with  the perception on school rule and regulation in promoting good behavior as the value of P-Value is greater than 0.05. All the other categorical explanatory variables present statistically significant  relationship-dependence with perception on school rule and regulation  in promoting good behavior (P-Value is lower than 0.05).

Author Response

Some responses for comments 

do all chi-square test in your table have the dof?

answer

no because degrees of freedom for chi-square is based on number of colomun and rows of contigency table i.e (C-1)*(R-1)

Table 1 have been drived

for example (937/2520)*449 =166.95 = 167 (roundup) (Mathematical rule concept, .95>=.5 =1)

for the others it has improved and there was correction taken 

Pearson Chi-square has discussed on methodology part

Father education level percentage is corrected

all chi-square results are discissed 

thus

participating in school rules formulation or implementation, father employment status and punishment encourages following rules and regulations are included

for all the significant level is 5% 

Generally detail comments were incorporated in the manuscript 
